# The effect of nature exposure on pain experience and quality of life in patients with chronic pain: A systematic review and meta-analysis protocol

**Matthew J. Lee**[1]*, **Aishwarya Pradeep**[2], **Katie Lobner**[3], **Oluwakemi Badaki-Makun**[4]*

**1** Hackensack Meridian School of Medicine, Nutley, New Jersey, United States of America, **2** Mayo Clinic Alix School of Medicine, Rochester, Minnesota, United States of America, **3** Johns Hopkins Welch Medical Library, Baltimore, Maryland, United States of America, **4** Johns Hopkins University School of Medicine, Baltimore, Maryland, United States of America

* matthew.lee@hmhn.org (MJL); kemi.badaki@jhmi.edu (OBM)

**Data Availability Statement:** No datasets were generated or analysed during the current study. Data extraction files from this study will be made available upon study completion.

## Abstract

### Background

Chronic pain is a complex condition with short and long-term effects on physical and psychosocial health. Nature exposure therapy has been investigated as a potential non-pharmacological intervention to improve physical and emotional health of individuals with chronic pain. This proposed systematic review aims to examine the effects of nature exposure therapy on pain experience and quality of life in patients with chronic pain.

### Methods

Studies will be identified by searching the MEDLINE, Embase and Cumulative Index for Nursing and Allied Health Literature (CINAHL) databases. All included studies will be required to be interventional controlled trials comparing nature exposure therapy to placebo or standard care in patients with chronic pain. Primary outcomes for this review will be pain intensity and quality of life scores. Secondary outcomes will include self-efficacy, depression and pain-related anxiety scores. If 2 or more studies are included, results will be pooled for meta-analysis. If meta-analysis is not possible, the results will be presented in a narrative form.

### Discussion

Given the adverse effects of opioid use, non-pharmacological interventions are a necessary alternative to treat patients with chronic pain. Nature exposure therapy is an intriguing example of such an intervention. We hope that this systematic review will guide future clinical decision-making for patients with chronic pain and provide evidence for or against the need for natural spaces and improved urban planning.

### Trial registration

PROSPERO registration number: CRD42021226949.

**Funding:** The authors received no specific funding for this work.

**Competing interests:** The authors have declared that no competing interests exist.

## Introduction

Chronic pain remains a complex public health issue with emotional, physical and socioeconomic implications [1–3]. According to the Centers for Disease Control and Prevention (CDC), approximately 20.4% of adults in the U.S experienced some form of chronic pain in 2019 and 7.4% reported it interfering with work and daily life activities [4]. While most attention is directed towards the adult population, a significant number of children develop chronic pain as well [5]. It is estimated that approximately 8% of children experience chronic pain worldwide [6]. For the purposes of this review, we will draw upon the International Association for the Study of Pain's (IASP) definition of pain as "an unpleasant sensory and emotional experience associated with actual or potential tissue damage." We will also use the ICD-11 definition of chronic pain as pain which has "persisted beyond normal healing time and has recurred for more than 3 months." Both primary chronic pain (i.e., headache pain, widespread pain, etc.) and chronic pain secondary to underlying disease (i.e., cancer-related pain, postsurgical pain, visceral pain) will be considered for this review.

Chronic pain can stem from several causes. Often it is the sequelae of persistent and recurring pain from chronic conditions, neuropathy or musculoskeletal causes such as joint pain [7]. Chronic pain can also be a potential long-term complication of surgical procedures. Although dependent on the type of operation, postsurgical chronic pain is estimated to affect up to 50% of adults following surgery [8].

Opioids have long been a popular treatment option in the management of patients with chronic pain [9]. However, given their addictive effects and high rates of opioid overdose deaths, non-pharmacological interventions have been examined as potential alternatives [10]. Nature exposure therapy is an example of such an intervention that may be useful in chronic pain rehabilitation.

Nature is defined as the animals, flora, landscape or natural products of the Earth that an individual can perceive or interact with through their senses. Exposure to nature has been linked to improvements in several health outcomes including pain intensity and anxiety [11–13]. While the mechanisms are still unclear, it has been hypothesized that green light from natural sceneries may induce positive emotions and subsequently alleviate pain and pain-associated psychological symptoms [14]. Through exposure to natural settings, nature exposure therapy utilizes these therapeutic effects of nature in the treatment of medical and psychiatric conditions.

Given the advancement of technology in recent years, nature exposure therapy may also integrate virtual components [15]. Virtual reality programs using natural scenery (e.g., forest, grasslands, ocean) and nature sounds have been used in medical rehabilitation and have demonstrated improvements in depression, stress and pain [16, 17]. With the lack of green spaces in certain urban communities, such interventions may be effective alternatives [18].

While the literature indicates that nature exposure therapy can have therapeutic effects, little is known about its impact in those with chronic pain. To date, there are no high-quality systematic reviews examining the effects of nature exposure therapy in patients with chronic pain. Thus, it is vital that this review investigates the literature and evaluates the potential benefits of this non-pharmacological intervention. This proposed systematic review aims to examine the effects of nature exposure therapy on pain experience and quality of life in patients with chronic pain.

## Methods

In conducting this systematic review, we will follow the methodology outlined in the Cochrane Handbook for Systematic Reviews of Interventions. This systematic review protocol is

reported in accordance with the Preferred Reporting Items for Systematic Reviews and Meta-Analysis (PRISMA-P) guide (S1 Appendix) [19] and has been registered in the International Prospective Register of Systematic Reviews (PROSPERO) (CRD42021226949). Upon completion, this systematic review will be reported in alignment with the PRISMA 2020 statement.

## Eligibility criteria

Controlled studies comparing nature exposure therapy to placebo or standard care in the treatment of patients with chronic pain will be included this study. We will consider studies with participants of all ages experiencing any primary or secondary chronic pain. Studies with interventions using physical or virtual nature exposure therapies to alleviate pain in either the inpatient or outpatient clinical environment will be included. Given the breadth of nature exposure, all ecosystems (e.g., greenspace, blue space) will be considered for this review. No limitations will be imposed on the form of nature exposure intervention, which can encompass both passive (e.g., observing nature) and active (e.g., participating in nature-based activities) experiences. The comparator group in this systematic review will be patients receiving placebo or standard pharmacological or non-pharmacological treatments for pain such as analgesics and physical therapy. There will be no restrictions placed on the year of publication and country of origin.

Trials using nature exposure therapies as secondary treatments or interventions involving the pharmacological use or consumption of medicinal plants, herbs and other flora will be excluded from this review. Trials using combination therapy of pharmacological analgesics and nature exposure as a primary intervention will also be excluded. Studies using subjective measures of pain as primary or secondary outcomes will not be considered for this review. Studies written in languages other than English will be excluded.

## Information sources and search strategy

To identify relevant studies, we will conduct a search of the MEDLINE, Embase, and Cumulative Index for Nursing and Allied Health Literature (CINAHL) databases. Our search strategies will utilize a combination of terms derived from the PICOS framework, as outlined in S1 Table and S2 Appendix.

## Selection of studies

Two review authors (MJL and AP) will first independently screen for study eligibility by reading the titles and abstracts. Studies inconsistent with the eligibility criteria will be excluded. Disagreements concerning study inclusion will be resolved through decision from a third author (OBM). Following title and abstract assessment, two independent authors (MJL and AP) will review the full text of the remaining eligible studies. Conflicts will be resolved by a third author (OBM). A PRISMA flow chart will be included in order to display the screening process.

## Data extraction and management

Two review authors (MJL and AP) will independently extract data using Covidence prior to entering information into Review Manager 5.4 (RevMan5.4). A third author (OBM) will adjudicate in case of disagreement. We will extract the following information:

- Bibliographic data (authors, years, date of publication)

- Methods: aims of study, study design, method of recruitment, declaration of interests of investigators

- Characteristics of participants (age, gender, condition, duration of pain, treatment)

- Description of nature intervention (virtual vs. physical, specific ecosystem, interactive vs. passive experience, duration of nature exposure)

- Description of control group intervention (pharmacological vs. non-pharmacological)

- Outcomes of Interest: (pain, quality of life and psychosocial outcomes)

## Outcome measures

**Primary outcomes.**   Reduction in pain intensity will be a primary outcome. We will examine quantitative pain intensity scores and self-report measures such as the Visual Analog Scale (VAS) and the numerical rating pain scale. Quality of life will also be a primary outcome that will be assessed. Quantitative measures such as the Quality-of-Life Scale (QLOS) and the 36-item short form survey (SF-36) will be used along with other measures.

**Secondary outcomes.**   Psychosocial outcomes will also be measured. Self-efficacy, depression and pain-related anxiety scores are secondary outcomes that will be assessed through this systematic review.

## Assessment of risk of bias in included studies

Two authors (MJL and AP) will independently assess the risk in accordance with the Cochrane bias risk tool [20]. We will assess the following items: blinding (participants, personnel, and outcome assessors); sequence generation; incomplete outcome data; and other potential threats to validity. We will assess the studies and judge each as: low risk, high risk or unclear risk of bias. We will make necessary judgements and a third author (OBM) will adjudicate in case of any disagreements.

## Data analysis and synthesis

Our decision to meta-analyze study data will be dependent on the number of included studies (2 or more) and whether trials are similar in terms of participants, settings, intervention, comparison and outcome measures. If meta-analysis is conducted, studies will be pooled using the RevMan5.4 software, utilizing the inverse variance method. We aim to apply a random-effect model as we anticipate our outcome measures to assess the same underlying intervention effect.

In order to account for differing outcome measurements across included studies, effect sizes will be presented using the standardized mean difference (SMD) with corresponding 95% confidence intervals. We anticipate that outcomes will be continuous and plan to evaluate differences in pre- and post-intervention values or absolute post-intervention values. Skewness of outcome data will be assessed using statistical methods. If substantial skewness is identified, it will be considered in our analysis and interpretation of results.

We will utilize the $I^2$ statistic to determine the extent to which variation across the included studies is due to heterogeneity compared to chance. We will interpret the percent ranges as follows: 0% to 40% suggests insignificant heterogeneity, 30% to 60% suggests moderate heterogeneity, 50% to 90% suggests substantial heterogeneity, 75% to 100% suggests considerable heterogeneity.

Subgroup analyses will be conducted if there is sufficient data available to compare different forms of nature exposure interventions (e.g., interactive vs. passive experience, physical vs. virtual, different ecosystems), as well as the duration of the nature exposure intervention and the duration of follow-up. If the review yields other significant findings, additional subgroup analyses may be carried out.

Sensitivity analysis will be conducted to evaluate the impact of excluding studies with high risk of bias. We will also use the Egger test and construct funnel plots to visually present the effect sizes and determine the presence of publication bias [21]. If meta-analysis is not possible, the results from the included studies will be presented in a narrative manner and findings within and between trials will be described and presented through figures and tables.

## Assessing the certainty of evidence

We will adhere to the Grading of Recommendations, Assessment, Development and Evaluation (GRADE) approach when evaluating the certainty of evidence. We will use the following criteria:

- High: very confident that true effect is close to estimated effect

- Moderate: moderately confident in estimated effect and true effect is likely close to that but may be different

- Low: confident in estimated effect is limited

- Very low: very little confidence in estimated effect

We will then create a "Summary of Findings" table using the GRADEpro software [22] and use the GRADE system to evaluate the evidence.

## Discussion

This systematic review aims to identify the comprehensive impacts of nature exposure therapy in patients experiencing chronic pain. While pain intensity is often the primary area of concern in patients with this condition, individual quality of life and psychosocial health are often affected as well. The literature suggests that the therapeutic effects of nature exposure therapy may improve these outcomes, making it an intriguing intervention in the rehabilitation of chronic pain. The results of this review will summarize the evidence and determine whether nature exposure therapy can have beneficial effects in patients with this condition.

To our knowledge, no systematic review currently exists that examines the effects of nature exposure therapy in patients with chronic pain. There is also a lack of universally recognized standards accessible to physicians for prescribing nature exposure as a means of treating chronic pain. We hope this review will serve as a significant resource to clinicians by providing objective evidence on a potential non-pharmacological alternative to opioids and other analgesics. Due to the wide range of nature exposure therapies, we aim to offer recommendations on effective types and durations of interventions for individuals with chronic pain. We believe our review will highlight the importance of conducting more high-quality randomized controlled trials on nature exposure therapy and increasing the availability of natural spaces in urban areas to support the management of chronic pain.

## Supporting information

**S1 Table. Search strategy.**
(TIFF)

**S1 Appendix. PRISMA-P checklist.**
(DOC)

**S2 Appendix. Search strategy.**
(DOCX)

## Author Contributions

**Conceptualization:** Matthew J. Lee, Oluwakemi Badaki-Makun.

**Data curation:** Aishwarya Pradeep.

**Investigation:** Matthew J. Lee.

**Methodology:** Matthew J. Lee, Aishwarya Pradeep, Katie Lobner, Oluwakemi Badaki-Makun.

**Software:** Matthew J. Lee.

**Supervision:** Oluwakemi Badaki-Makun.

**Visualization:** Matthew J. Lee.

**Writing – original draft:** Matthew J. Lee.

**Writing – review & editing:** Matthew J. Lee, Aishwarya Pradeep, Katie Lobner, Oluwakemi Badaki-Makun.

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
