## [Decision Letter · Decision Letter 0]

15 Nov 2022

PONE-D-22-27571The Effect of Nature Exposure on Pain Experience and Quality of Life in Patients with Chronic Pain: A Systematic Review and Meta-Analysis ProtocolPLOS ONE

Dear Dr. Lee,

Thank you for submitting your manuscript to PLOS ONE. After careful consideration, we feel that it has merit but does not fully meet PLOS ONE’s publication criteria as it currently stands. Therefore, we invite you to submit a revised version of the manuscript that addresses the points raised during the review process.

We look forward to receiving your revised manuscript.

Kind regards,

Germain Honvo, Ph.D.

Academic Editor

PLOS ONE

Journal Requirements:

2. We note that this manuscript is a systematic review or meta-analysis; our author guidelines therefore require that you use PRISMA guidance to help improve reporting quality of this type of study. Please upload copies of the completed PRISMA checklist as Supporting Information with a file name “PRISMA checklist”.

**Additional Editor Comments:**

There are some important methodological flaws in this protocol, which make it not acceptable for publication in its current form. Here are some specific issues that need to be addressed.

1. Please note that there is a difference between methodology guidelines and reporting guidelines. The PRISMA-P is rather a reporting guideline. The authors should refer to adequate methodology guidelines for the conduct of this systematic review (in this particular case, the guidelines in the Cochrane handbook may be appropriate), then to the PRISMA-P as a reporting guideline for this protocol manuscript. Therefore, the sentence “This systematic review protocol is designed in accordance with the guidelines provided in the Preferred Reporting Items for Systematic Reviews and Meta-Analysis Protocols (PRISMA-P) statement” should be amended.

2. Please note that exclusion criteria are not opposite of inclusion criteria. Therefore, the following should be removed from the manuscript: “Case control, proof-of-concept, two-gate studies and case studies will be excluded”; since this systematic review is on interventional controlled trials.

3. The authors need to provide adequate and detailed search strategy for at least one of the databases considered for this systematic review. This should be provided as a supplementary file.

4. The authors declared that “The search strategy will be composed of four concepts combined with Boolean operators: “nature,” “chronic pain,” “treatment,” and “controlled trials.” Considering the PICOS framework and the research question in this systematic review, the ‘concept’ “treatment” does not seem to be the most adequate in the search strategy to denote the ‘outcome’.

5. The Editor does not see the added value of a blank flow-chart in a protocol manuscript, though this practice has been sometimes seen in the literature. Please consider removing this.

6. Instead of presenting the following subtitles in the methods section (i.e., ‘Types of Studies’, ‘Types of Participants ‘, ‘Types of Intervention’) and in accordance with the PRISMA-P checklist, the Editor advises the authors to report the ‘Eligibility criteria’, as a clear section, with one paragraph on inclusion criteria, and another one on exclusion criteria. Please consider being concise. Then, another separate section on ‘Outcomes and prioritization’ should be reported.

7. The following should be removed from the data extraction section: “Risk of bias: specified under Assessment of risk bias in included studies”. This is not a data to extract.

8. For the structure of the methods section, please consider strictly following the PRISMA-P. This will ease reading and understanding for potential future readers of this manuscript.

9. For risk of bias assessment, instead of stating “… will independently assess the risk in accordance with the Cochrane Handbook for Systematic Reviews of Interventions”, please state that the Cochrane risk of bias tool will be used (not the handbook).

10. In accordance with the PRISMA-P, please remove the GRADE assessment from the data analysis section. These are indeed two different things.

11. The Data synthesis section is very poorly described. Please refer to previous well reported meta-analyses and to the Cochrane handbook for systematic reviews for guidance. The statistical analyses planned should be clearly reported in this section, with assumptions guiding the choices made.

12. Please remove the section on “Measures of treatment effect” from the manuscript and move relevant information to the data analysis section. Please note that the following sentence in this section does not make any sense: “To test the significance of the RR and 95% CI, we will use the chi squared test”.

13. What is the meaning of this sentence in a meta-analysis where the authors are planning the calculate SMDs: “T-tests and ANOVA tests will be utilized when comparing two means and multiple mean values respectively”?

14. The Editor advises the authors to take the necessary time to adequately revise this manuscript before resubmitting it to the Journal. If the authors feel it necessary, the Editor advises them to ask for the help of colleagues with strong experience in the conduct of systematic reviews and meta-analyses, to improve this protocol (indeed, the Editor thinks this may be necessary).

Thank you.

Reviewers' comments:

Reviewer's Responses to Questions

**Comments to the Author**

1. Does the manuscript provide a valid rationale for the proposed study, with clearly identified and justified research questions?

Reviewer #1: Yes

Reviewer #2: Yes

2. Is the protocol technically sound and planned in a manner that will lead to a meaningful outcome and allow testing the stated hypotheses?

Reviewer #1: Yes

Reviewer #2: No

3. Is the methodology feasible and described in sufficient detail to allow the work to be replicable?

Reviewer #1: No

Reviewer #2: Yes

4. Have the authors described where all data underlying the findings will be made available when the study is complete?

Reviewer #1: Yes

Reviewer #2: Yes

5. Is the manuscript presented in an intelligible fashion and written in standard English?

Reviewer #1: Yes

Reviewer #2: Yes

6. Review Comments to the Author

Reviewer #1: 

This review and meta-analysis will examine the effects of nature exposure (both actual and visual) on pain and quality of life measures among individuals with chronic pain. This review is timely and will be of interest to the readers from various disciplines. Given the high percentage of pain conditions in the general population, as well as the detrimental effects of common pain care approaches (e.g., opioid prescriptions), it is critical to develop non-pharmacological options for patients with pain and anxiety about their pain. I suggest consideration of the following minor points to strengthen the methods of this review and meta-analysis:

• “Greenspace” should be added to search terms in addition to just “nature,” especially given greenspace and the color green may have explicit pain reducing properties (perhaps more so than other types of nature, however that remains to be seen).

• While this may be the intentions of the authors already, it would help to explicitly state in the data extraction more overt details in addition to “type” of nature. Does this mean virtual or actual nature exposure? Or does this mean various nature ecosystems such as greenspace versus bluespace or others? There are also active versus passive experiences with nature, does this matter? How much time of exposure would also be very helpful, as there may be a minimum threshold. Just a few more explicit details on what the authors intend to collect would be very helpful. A couple of these thoughts came to the fore in the discussion, however should be listed explicitly in the methods.

• The above point will be a critical component of the discussion of this review. The implications being that perhaps some nature exposure might be beneficial, but there might be a minimum threshold that is needed to reach that point (e.g., minutes spent viewing nature, “quality” of nature viewed). From this exciting review, I hope that the authors are able to glean specific recommendations moving forward for both future research and practice. However if the actual intervention of nature (i.e., the independent variable in this case), is not documented extensively, the review may fall short of this.

• May also be worth noting that nature prescriptions from physicians are gaining traction in several countries, but specific guidelines for these prescriptions for any conditions (including pain) are not well fleshed out, and this review will be a critical step in synthesizing the research to reach “prescription guidelines.”

Reviewer #2: 

In general, the protocol regarding a systematic review and meta-analysis of the effect of nature exposure on pain experience and quality of life in patients with chronic pain was well written. However, no control or comparison was described in the PICO framework. Besides, inclusion and exclusion criteria of the articles were not described in detail.

---

## [Author Response · Author response to Decision Letter 0]

28 Mar 2023

Our response to the reviewers and editor is provided as an attachment

---

## [Decision Letter · Decision Letter 1]

16 Jun 2023

PONE-D-22-27571R1The Effect of Nature Exposure on Pain Experience and Quality of Life in Patients with Chronic Pain: A Systematic Review and Meta-Analysis ProtocolPLOS ONE

Dear Dr. Lee,

Thank you for submitting your manuscript to PLOS ONE. After careful consideration, we feel that it has merit but does not fully meet PLOS ONE’s publication criteria as it currently stands. Therefore, we invite you to submit a revised version of the manuscript that addresses the points raised during the review process.

We look forward to receiving your revised manuscript.

Kind regards,

Germain Honvo, Ph.D.

Academic Editor

PLOS ONE

Journal Requirements:

Additional Editor Comments:

Thank you to the authors for their revisions, which have significantly improved this systematic review protocol. Please consider some few (important) additional comments to further improve this protocol.

1) Abstract: Please remove the following from the abstract, as per a previous Editor’s comment: “Case control, proof-of-concept, two-gate studies and case studies will be excluded” (lines 32-33).

2) Introduction to the methods section: Please consider reporting first which methodology guideline will be followed to conduct this systematic review before stating which one was followed to report the current protocol. Please refer to the Editor’s previous comment n°1 (first peer review round). In addition to these two guidelines, please also add that once completed, this systematic review will be reported according to the PRISMA 2020 statement.

3) Eligibility criteria: Please move the exclusion criterion “Studies written in languages other than English will be excluded” to the end of the list of exclusion criteria, as this is not the most important one in this study.

4) Section on “Search strategy”: Please remove the sentence starting by “To ensure consistency and collaboration between…” (lines 120-122), as this has nothing to do with search strategies.

5) Please consider providing the search strategies as a supplementary MS Word document. Second, as Medline and Embase do not use the same thesaurus, please note that it is necessary to provide search strategies tailored to each database separately (including CINAHL). Please note that the validity of systematic reviews primarily depends on the quality of search strategies.

6) Selection of studies: Please change “Studies inconsistent with the inclusion criteria will be excluded” to “Studies inconsistent with the eligibility criteria will be excluded”. Please note that both inclusion and exclusion criteria are always considered for studies selection (including title/abstract selection). The last but one sentence (lines 129-131) is not very clear. Please rephrase or delete; then add that full text selection will follow title/abstract selection.

7) Data analysis: For effect sizes calculations, anticipating that different outcome measurements may have been used, please consider using SMDs only (not MD and SMD). This will ease comparisons of treatment effects.

Please note that subgroup analysis is heterogeneity assessment.

Please remove “If >10 studies are included in meta-analysis, we will assess for heterogeneity”.

Please refer to the Cochrane handbook and to high quality systematic reviews for guidance to improve this section. Thank you for your understanding.

8) Please consider changing the title ‘Assessing certainty in findings’ to ‘Assessing the certainly of evidence’.

Reviewers' comments:

Reviewer's Responses to Questions

**Comments to the Author**

1. Does the manuscript provide a valid rationale for the proposed study, with clearly identified and justified research questions?

Reviewer #2: Yes

2. Is the protocol technically sound and planned in a manner that will lead to a meaningful outcome and allow testing the stated hypotheses?

Reviewer #2: Yes

3. Is the methodology feasible and described in sufficient detail to allow the work to be replicable?

Reviewer #2: Yes

4. Have the authors described where all data underlying the findings will be made available when the study is complete?

Reviewer #2: No

5. Is the manuscript presented in an intelligible fashion and written in standard English?

Reviewer #2: Yes

6. Review Comments to the Author

You may also provide optional suggestions and comments to authors that they might find helpful in planning their study.

Reviewer #2: I can confirm that my previous comments have been adequately addressed. I have no further comments at this time. Thank you.

7. PLOS authors have the option to publish the peer review history of their article (what does this mean?). If published, this will include your full peer review and any attached files.

Reviewer #2: No

---

## [Editor Report · Decision Letter 2]

22 Aug 2023

The Effect of Nature Exposure on Pain Experience and Quality of Life in Patients with Chronic Pain: A Systematic Review and Meta-Analysis Protocol

PONE-D-22-27571R2

Dear Dr. Lee,

We’re pleased to inform you that your manuscript has been judged scientifically suitable for publication and will be formally accepted for publication once it meets all outstanding technical requirements.

Kind regards,

Germain Honvo, Ph.D.

Academic Editor

PLOS ONE

Additional Editor Comments (optional):

Final minor comments:

Thank you for addressing all the previous comments. Please make these few additional changes before publication of your protocol.

- Methods section: Please change the title « Search strategy » to « Information sources and Search strategy”, to best fit with the content of this section.

- In the data analysis section, are you mentioning “histograms” or “funnel plot”? Please consider revising this before publication of the protocol. Please refer to well reported systematic reviews and meta-analyses. Thank you.

- The Editor suggests applying a random-effects model, instead of a fixed-effect model (considering what you stated in the rest of the data-analysis section). Therefore, the last sentence of the first paragraph of the “Data analysis and synthesis” should be rephrased. Please consider revising this before publication of the protocol. Thank you.
---

## [Editor Report · Acceptance letter]

19 Sep 2023

PONE-D-22-27571R2 

The Effect of Nature Exposure on Pain Experience and Quality of Life in Patients with Chronic Pain: A Systematic Review and Meta-Analysis Protocol 

Dear Dr. Lee:

I'm pleased to inform you that your manuscript has been deemed suitable for publication in PLOS ONE. Congratulations! Your manuscript is now with our production department. 

Kind regards, 

on behalf of

Dr. Germain Honvo 

Academic Editor

PLOS ONE